# Efficient electrospray deposition of surfaces smaller than the spray plume

Sarah H. Park [1], Lin Lei[2], Darrel D'Souza[2], Robert Zipkin [3], Emily T. DiMartini [4], Maria Atzampou[4], Emran O. Lallow[2], Jerry W. Shan [2], Jeffrey D. Zahn [4], David I. Shreiber[4], Hao Lin [2], Joel N. Maslow[5] & Jonathan P. Singer [1,2] ✉

Electrospray deposition (ESD) is a promising technique for depositing micro-/nano-scale droplets and particles with high quality and repeatability. It is particularly attractive for surface coating of costly and delicate biomaterials and bioactive compounds. While high efficiency of ESD has only been successfully demonstrated for spraying surfaces larger than the spray plume, this work extends its utility to smaller surfaces. It is shown that by architecting the local "charge landscape", ESD coatings of surfaces smaller than plume size can be achieved. Efficiency approaching 100% is demonstrated with multiple model materials, including biocompatible polymers, proteins, and bioactive small molecules, on both flat and microneedle array targets. UV-visible spectroscopy and high-performance liquid chromatography measurements validate the high efficiency and quality of the sprayed material. Here, we show how this process is an efficient and more competitive alternative to other conformal coating mechanisms, such as dip coating or inkjet printing, for micro-engineered applications.

Efficient surface deposition of chemical and biological materials is a process of broad significance in many applications including transdermal drug delivery, biosensing, tissue engineering, cell stimulation, and wound healing, among others[1–4]. Similar to the electrostatic spray technique used in the automotive and agricultural industries to coat large surfaces[5], ESD has emerged as a promising contender in the manufacturing space of bio-functional surfaces[6]. In this technique, a high electric field drives surface instabilities from a solution reservoir to produce monodisperse, charged droplets which emerge from a needle tip and move toward a grounded target (Fig. 1a)[7–9]. Highly controllable droplets as small as 100 nm in diameter can be quickly produced via ESD, which makes the technique particularly attractive where micro- and nano-scale coatings and particles are needed. However, for templates and targets smaller than the characteristic spray plume, the deposition efficiencies reported for these sprays still indicate significant materials loss[10–12]. In this work, we demonstrate

that by manipulating the collective "charge landscape" made up of the external field and accumulated charges, we can achieve ESD with 100% efficiency using near-field techniques on targets that are significantly smaller than the plume. In addition to highly targeted and efficient coatings, our small therapeutic molecule demonstrated no decomposition post-processing via high-performance liquid chromatography measurements (HPLC). Using microneedle arrays (MNAs) as a model target for this study, we anticipate our work using ESD coatings to expand towards therapeutics delivery methods for costly bioactive materials and scale-up in manufacturing.

ESD has been utilized to process nanoparticles (NPs) in biomedical applications, examples of which include ESD of chitosan and collagen as drug delivery carriers[13,14], polycaprolactone NPs layers to control alignment and pattern growth for cell culture[15], and glass coatings onto metallic implants for bonding with native bone and osseointegration[16,17]. In addition, ESD can safely deposit protein NPs

[1]Department of Materials Science and Engineering, Rutgers, The State University of New Jersey, Piscataway, NJ 08854, USA. [2]Department of Mechanical and Aerospace Engineering, Rutgers, The State University of New Jersey, Piscataway, NJ 08854, USA. [3]MedChem 101 LLC, Plymouth Meeting, PA, USA. [4]Department of Biomedical Engineering, Rutgers, The State University of New Jersey, Piscataway, NJ 08854, USA. [5]GeneOne Life Science, Seoul, South Korea. ✉e-mail: jonathan.singer@rutgers.edu

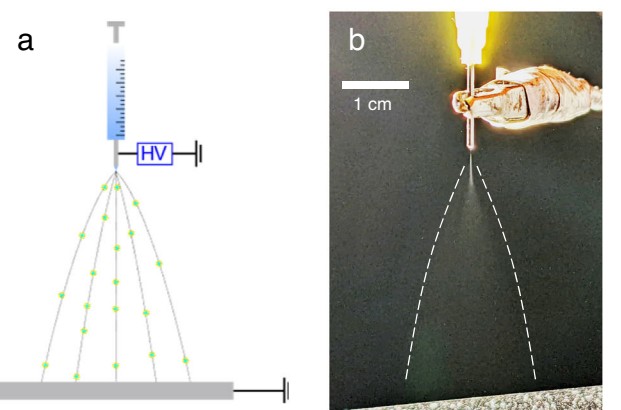

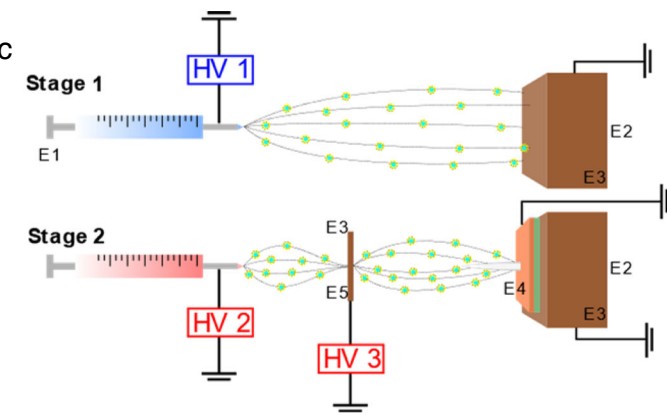

**Fig. 1 | Schematic of the electrospray setup. a** Schematic and **b** photograph of an ESD experiment where a spray plume directed at a grounded target is generated from a solution reservoir held at high voltage. White dashed lines are provided as a guide to the eye of the plume. **c** Schematic of the spray system and process, highlighting different enhancements (denoted 'EX'). In stage 1, a negative polarity ethanol spray (E1) is sprayed directly at a large extractor ground (E2) which is coated in insulating Kapton tape (E3). While the focus ring is in place during this treatment, no clip is applied and thus it is not electrified or grounded. Then in stage 2, a grounded target with an insulating mask (E4) is placed on the extractor ground. It is then sprayed by the spray solution at positive polarity which is stabilized by a focus ring (E5). The ring, and all other proximate metal surfaces, is also coated in insulating tape.

that maintain bioactivity after spraying[12,18]. In these and other applications, ESD demonstrates high efficiency in a regime where the template size is, in general, greater than the plume size which is commensurate to the spray distance of 4 cm in this study (Fig. 1b). This is due to the fact that the non-inertial droplets follow the field lines to their grounded target[8,19], eliminating almost all droplets missing the target, termed "over-spray."

The regime where the template/target size is smaller than the plume is much less explored and yet is of great importance and relevance to emerging applications, such as MNA-based drug delivery and the creation of functionalized sensor electrodes[20,21]. Such sprays reduce the mechanical precision required of near-field techniques (e.g., inkjet printing or dip coating), but efficiency in these sprays has been studied only limitedly. Morozov and Morozova reported up to 80% efficiency and 55% protein activity of alkaline phosphatase sprayed at a distance of 10 mm at a 20 mm² target through a 3 mm² insulating, non-contacting mask[12]. Work of Kingsley et al. showed deposition efficiency to be less than 7% at a distance of 55 or 65 mm onto a 36 mm² target[10]. All additional material ostensibly found other grounded targets in their spray chamber. In another study, Angkawinitwong et al. experimented with the coating of MNAs on a 1 cm² electrode located 10 cm from the deposition needle. Deposition efficiencies of less than 30% were reported for these substrates[11]. To overcome this challenge, the electric field will have to be engineered, e.g., with a guide ring to focus the field lines on the small target[12,22–26]. Charge accumulation from the arriving spray on either the target or the insulated area modifies field distribution, thus further limiting spray[27,28]. While this limiting can be exploited to conformally coat complex surfaces in the self-limiting electrospray deposition (SLED) mode[7,9,29], the charge buildup may eventually compromise the stable cone-jet mode necessary for ESD[19].

In this work, we examine three model geometries: MNA with a surface area of approximately 0.2 mm² per needle, a silicon chip, and an electrode test pattern printed on Borofloat glass. The total surface area of the MNA, silicon chip, and test pattern are approximately 3 mm², 1 cm², and 0.9 cm², respectively. The MNA represents a general 3-D surface and is a platform that demonstrated great advantages in transdermal drug and vaccine delivery[30–34], including dose-sparing effects and room-temperature stability to eliminate cold-chain requirements[35]; whereas the silicon chip and test pattern is a generic 2-D example representing any small and flat substrate. We focus on biologically relevant materials, including trehalose, GLS-1027 (a

therapeutic small molecule), poly(ethylene glycol) diacrylate (PEGDA), polyvinylpyrrolidone (PVP), poly(3,4-ethylenedioxythiophene) (P3KT), polystyrene sulfonate (PEDOT:PSS), GLS-6150 (a hepatitis C virus DNA vaccine)[36], trehalose-stabilized horse radish peroxidase (HRP, a protein complex), and bovine serum albumin (BSA) protein. In addition to efficiency, we also validate functional activity or structural fidelity of the GLS-1027, GLS-6150, and HRP. Our work establishes ESD as a viable alternative to dip coating, which requires a standing reservoir of unused material, and inkjet printing, which is a serial process requiring expensive positioning equipment.

## Results
### Engineering the charge landscape
As mentioned, for targets smaller than the characteristic spray plume, the charge landscape can be modified to optimize the material deposition efficiency. The general hypothesis (shown schematically in Fig. 1c) is that efficient sprays should: (1) start from a relatively "blank state" and not be affected by previous sprays; (2) be free from alternative targets; (3) employ a focus ring; and (4) have a large extractor ground placed behind the target from the perspective of the spray. In addition, to maximize the effects of the deposited charge and minimize any effects that occurred after deposition, humidity was kept low for all sprays which has previously been shown to amplify self-limiting effects in ESD[37]. Experimental parameters for all sprays are listed in Supplementary Table 1. We systematically investigated five distinct enhancements to optimize the charge landscape of electrospray for deposition efficiency on an MNA (Target 1; T1), as well as a flat silicon chip (T2) and an electrode test pattern on Borofloat glass (T3): negative-polarity ethanol pre-spray to eliminate residual charges (E1); a large, grounded extractor beneath the target (E2); insulating tape on all non-target metal surfaces (E3); an insulating mask on unwanted regions of the target (e.g.: exposed electrodes on the MNA) (E4); and a focus ring to narrow the plume (E5). Ultraviolet-visible spectroscopy (UV-vis) efficiency results of trehalose ESD coatings deposited with and without the specific efficiency enhancement strategies for MNAs are shown in Fig. 2a where rhodamine B was used as a tracer molecule (see Supplementary Fig. 1 for calibration curve). As will be discussed, this approach was validated through several other quantitative metrics. Results with all enhancements for the flat silicon chip and test pattern are also shown, with both geometries used as a target to demonstrate that ESD is not target-specific and can be generalized. While T2 achieved a deposition efficiency of 110 ± 25%, T3 achieved a deposition

efficiency statistically similar (*P*-value = 0.9471) but slightly less than that at 96 ± 18%. For T3, the target was grounded such that the ground was in the spray path. Thus, the decrease in efficiency was likely caused by the ground both collecting and removing sprayed material from the target. Since trehalose is not a self-limiting material, deposition on T3 occurred as a mounded spot, albeit one that did not over-spray the surrounding substrate, and would not be ideal for electronic coatings. In contrast, Supplementary Fig. 2 demonstrates how a self-limiting material, PVP, can achieve a qualitatively more uniform and more tunable final thickness, as we demonstrated in further detail recently for another material, polystyrene, sprayed in SLED and non-SLED conditions[38]. In SLED, the material arrives onto a target as a relatively dry spray. Because the droplets carry charge, repulsive effects due to accumulation of charge in a coating may be expected to reduce the efficiency of the approach, as the charge eventually begins to repel itself over time. This is at times beneficial, since the charged spray is redirected to regions that are uncoated such that manipulation of the electrostatic repulsion, hydrodynamic forces, and evaporation kinetics can be employed to conformally cover 3-D architectures with micro-coatings possessing either nano-shell, nanoparticle, or nanowire microstructures[7]. Regions that were damaged in the pattern, and thus did not have a conductive path, also led to no deposition in both materials, illustrating the high selectivity of ESD. Further optimization of the in-plane coating uniformity will be left to future work. All targets and materials were sprayed at a constant spray time of 30 min, but Supplementary Fig. 3 demonstrates that achieving high deposition efficiencies can be done at various dosages for both self-limiting and non-self-limiting materials. This additionally illustrates the facile means of controlling needle dose in MNAs via ESD, which can be accomplished continuously through deposition time.

With all enhancements, the deposition efficiency of trehalose sprays coating on both MNAs and the silicon chip was approaching 100%. We hypothesize that the slightly greater than 100% apparent efficiency of the coating on individual sprays is due to the combined effects of uncertainties due to the small amounts of material and the accumulation of some dried material on the tip of the needle between spraying targets as well as during the stabilization of the spray. From an ultimate system-design standpoint, automated sample motion and spray stabilization, such as recently shown by Toth et al. [39], would likely improve the precision of the technique. Although we anticipate employing a similar design in the future to control spray stabilization, our work has shown that there are still significant differences between the enhanced and unenhanced results without the use of such system design.

The negative pre-spray (-E1) establishes the "blank slate" condition, without which the charge gradually builds up in the chamber to the point of influencing and destabilizing the spray. Allowing the charge to accumulate in the spray chamber resulted in a more variable spray efficiency of 60 ± 46%. Removing the extractor ground (-E2) decreased efficiency to 55 ± 32% which arises from charge screening and an eventual destabilization of the spray, as will be discussed further below. Most significant—and intuitive—is the effect of providing additional conductive surfaces for the spray to target. This is shown here in two ways. The first is to remove the masking tape from the extractor ground and focus ring (-E3, Fig. 2d). Upon doing so, the extractor ground and focus ring provide for a much larger conductive surface for the spray to deposit. The role of the masking tape is to initially build charge during the first few moments of spray to redirect the field lines towards the unmasked portions of the target. The apparent deposition efficiency is 9 ± 2%, indicating that approximately 90% of the spray is diverted to these large alternative targets, particularly the ring, which does not have a field of its own strong enough to repel the spray and requires a slight build-up of charge. The second is removing the silicone target mask (-E4, Fig. 2c), which allows for 55 ± 35% of the coating to be deposited on unwanted portions of the substrate, consistent with the quantity of non-desired conductive surface. The addition of the focus ring homogenizes the electrostatic field in the interelectrode space to redirect the spray to the desired grounded substrate. Removal of the focus ring (-E5) allows droplets to follow a wider range of field paths and reduces efficiency to 21 ± 15%. As previously noted, implementing all these enhancements (Fig. 2e) leads to 104 ± 10% of the spray arriving at the needle tips. Supplementary Movie 1 of an enhanced spray shows no excess deposition or spray instability. Shifting to a silicon chip (Fig. 2f) maintains essentially 100% efficiency (110 ± 25%).

While all these results are relatively intuitive, the role of the extractor ground is perhaps the most subtle. As shown when the tape is removed (-E3), the spray is initially directed to all conductive surfaces in the chamber, including the positively biased focus ring. Early in the spray, charge is therefore deposited on all of the accessible conductive surfaces, even if they are coated with insulating tape or masked, as has been observed in near-field ESD templating[26]. These charges create their own field which opposes the ESD field. If this counter field is too strong, the spray will be completely destabilized. Even barring this outcome, if the charge on the insulated surfaces is too large, it will screen the relatively small grounded target surface. This hypothesis was tested through a simplified finite element method (FEM) model. In this model, the geometry was reduced to 2-D axisymmetric with the

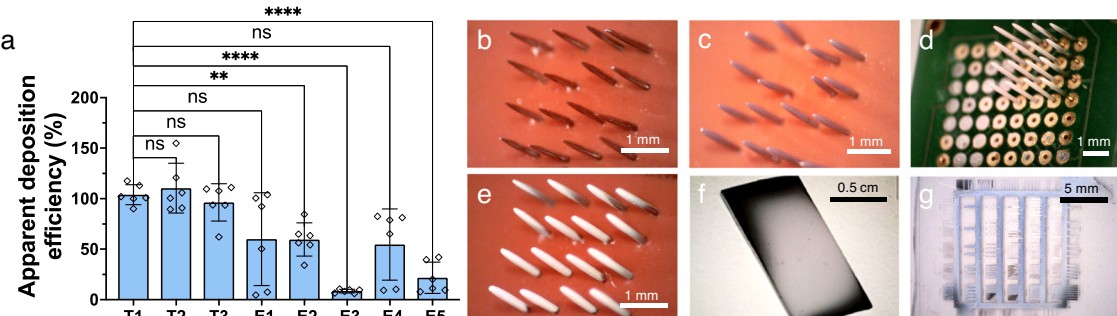

**Fig. 2 | Resulting deposition efficiencies with and without enhancements. a** Deposition efficiency for trehalose spray coatings with all enhancements applied on an MNA (T1), a silicon chip (T2), and an electrode test pattern on a Borofloat glass chip (T3) where "T" represents "targets". MNA substrates with a single enhancement removed are labeled as such: no negative-polarity ethanol pre-spray (-E1), no grounded extractor (-E2), no insulation on focus ring or extractor (-E3), no insulating silicone mask on the target (-E4), and no focus ring (-E5) where "-EX" represents "removing enhancement X with others in place".

Pairwise comparisons were conducted using ANOVA multiple comparisons tests. For all samples, *n* = 6. Error bars indicate standard deviation. Each point indicates a single measurement. ns = not significant; *$p$ = 0.0213 (T1 vs. -E2) and 0.0201 (T1 vs. -E3); **$p$ = 0.0079; ****$p$ < 0.0001. Photographs of: **b** bare MNA, **c** trehalose-sprayed MNA without insulation on focus ring or extractor, **d** unmasked trehalose-sprayed MNA, **e** trehalose-sprayed MNA with all enhancements, **f** trehalose sprayed silicon chip, and **g** trehalose sprayed onto a microelectrode test pattern.

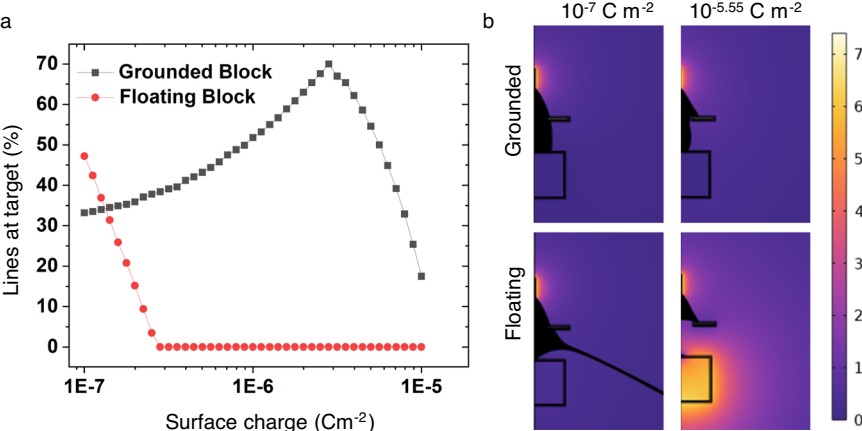

**Fig. 3 | Simulation of the surface charge effects observed with and without the use of a grounded extractor. a** Field line traces for different surface charges for the grounded and floating extractors. **b** Simulation results for field lines terminating at the target surface with a grounded and floating extractor ground. Field potential color map units are kV.

target and extractor ground represented as cylinders (see details in Supplementary Fig. 4). The spray was conceptualized as following field lines emitted from the tip of the needle. If the field line terminated at the target, we may consider that any spray that atomizes onto the path of that specific field line will also arrive at the target with the density of field lines proportional to the deposition rate. To simulate the spray escaping into the chamber, the outer boundary was also modeled as grounded. Charge was then added to all insulated surfaces for two cases: (1) the extractor grounded, similar to T1, and (2) the extractor floating, similar to -E2. At low charge, the floating ground case has more field lines terminating at the target than the grounded case, due to spray being directed to the large ground (Fig. 3a). However, when the charge on the surfaces is increased, the target is rapidly screened in the floating case, redirecting spray to the ring and surroundings while the target receives more of the field lines in the grounded case. The fraction of field lines that terminate at the target as a function of surface charge is shown in Fig. 3b. While this illustrates the difference observed between the T1 and -E2 cases in Fig. 2a, it underestimates the effect due to assumptions in the simplified model. This simulation is oversimplified in that in actuality, the charge is not uniformly distributed. Indeed, the local density of field lines may be expected to be proportional to the instantaneous charge-deposition rate. While future work will develop a more sophisticated model that captures this behavior, this result still qualitatively supports the role of the extractor ground in stabilizing the field and directing it towards the grounded target.

**Spray efficiency of model materials**

To demonstrate effectiveness with a wide range of biologically-relevant materials, we have selected trehalose, a small molecule used as a matrix material; PEGDA, a hydrogel precursor material; PVP, a biocompatible glassy polymer; PEDOT:PSS, a water-dispersible conductive polymer[40]; and three biologically-active substances: GLS-1027, an immunomodulating small molecule[41]; GLS-6150, a plasmid DNA vaccine; and BSA, a protein, as model materials. These are used to demonstrate examples of materials that could be employed and are not exhaustive of either materials or categories of materials that can be applied using this technique. It should be noted that of these, the PVP, as a glassy monomer, is expected to satisfy the conditions for SLED[9] and, thereby, should rapidly accumulate repulsive charge. All materials were sprayed onto MNAs and measured using UV-vis for efficiency with the exception of PEDOT:PSS and BSA. PEDOT:PSS and BSA were sprayed onto 1 cm² silicon chips for ease of re-dispersion post-processing as electrostatic interactions can improve adhesion, thus

making it more difficult to re-disperse these molecules through complete immersion and sonication[40,42]. In the case of these latter materials, we also did not employ the rhodamine internal standard as PEDOT:PSS's coloration interfered with the rhodamine peak and so that we could confirm the BSA concentration through a Pierce assay. Resulting deposition efficiency results for the selected materials are shown in Fig. 4a where all materials have mean apparent efficiencies greater than 97% where UV-vis measurements are validated by various other methods of measurement as shown in Supplementary Fig. 5. In all cases, we assume that the composition of the spray remains constant from the syringe to the target. This assumption can be justified by (1) the use of relatively dilute solutions such that precipitation at the spray needle tip is unlikely and (2) based on the commensurate molecular weight of the tracer and the lightest payload (479 g mol$^{-1}$ for rhodamine as compared to 205 g mol$^{-1}$ for GLS-1027), there is roughly equal likelihood of atomization and diffusive or convective removal of material and tracer. In addition to apparent deposition efficiencies nearing 100%, the sprayed materials maintain functionality post-spray. When sprayed with 2,2-dimethoxy-2-phenylacetophenone (DMPA), a photoinitiator, the PEGDA coating can then be cured to form a hydrogel. This hydrogel coating can be seen in Fig. 5a, where the rhodamine fluorescence of an absorbed fluid in the hydrogel also demonstrates function post-processing.

Despite achieving efficiencies nearing 100% with PEDOT:PSS, our results yielded deposition efficiencies at $66 \pm 10\%$ for an alternative conductive polymer, poly[3-(potassium-6-hexanoate) thiophene-2, 5-diyl] (P3KT). While P3KT efficiencies are still higher than the previous studies reported, its deposition efficiency was likely reduced by solution compatibility. For the spray setup, P3KT was eluted in 1:4 water:ethanol, similar to the other materials used. The rhodamine tracer was not included, as P3KT has a quantifiable absorption peak. Although P3KT is somewhat water-soluble, it is not soluble in ethanol, and modifiers, such as ammonium hydroxide, have been shown to be necessary to enhance its aqueous solubility[43,44]. The relatively low solubility may have resulted in some agglomerates or condensation at the Taylor cone, impeding flow and introducing instabilities at the needle tip, resulting in lower deposition onto the target. A more favorable solvent may improve the stability of the spray cone to achieve a higher deposition efficiency.

**Trehalose assay results**

Using an enzymatic trehalose assay kit, the deposition of trehalose onto the MNAs was measured to validate the UV-vis efficiency approach. The efficiency results from the assay ($113 \pm 23\%$) were not

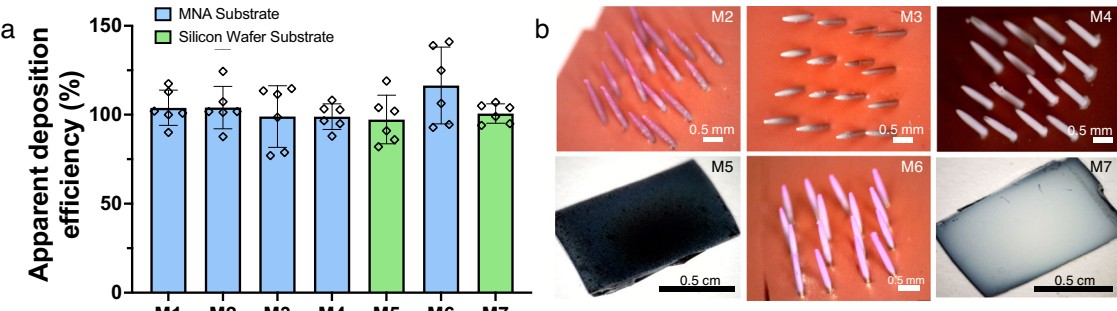

**Fig. 4 | Deposition efficiencies and images of the model active materials using all enhancements. a** Deposition efficiency for materials (denoted 'MX'): Trehalose (M1), GLS-1027 (M2), PEGDA (M3), PVP (M4), PEDOT:PSS (M5), GLS-6150 (M6), and BSA (M7). For all samples, $n = 6$. M5 and M7 were sprayed onto a 1 cm² silicon chip and quantified with a modified procedure. Error bars indicate standard deviation. Each point indicates a single measurement. **b** Needle and silicon chip images are shown and correspond to their respective labels. The coloration is due to the rhodamine tracer or, in the case of PEDOT:PSS, the color of the material.

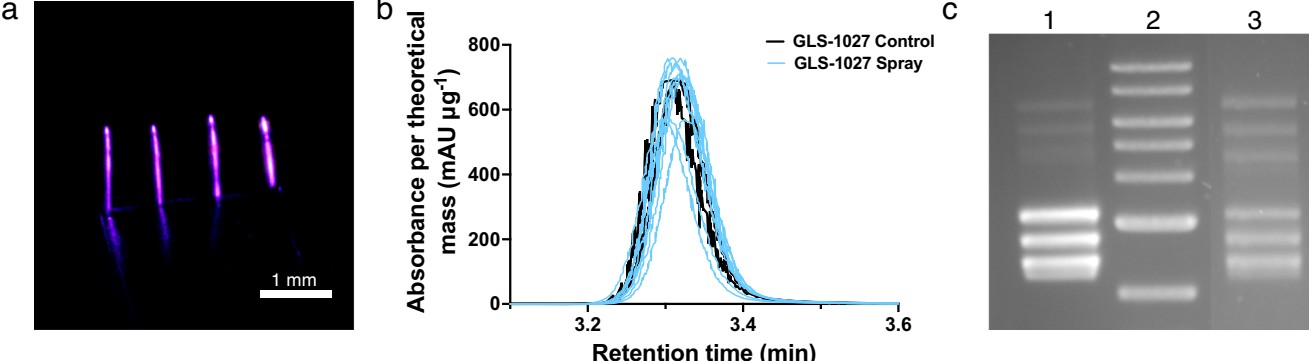

**Fig. 5 | Function and structure of model materials post-processing. a** Sprayed 1:0.036 PEGDA:DMPA coating. Hydrogel coating fluoresces after being dipped in water dyed with rhodamine B. **b** HPLC results of the sprayed GLS-1027 targets ($n = 3$) compared to the GLS-1027 control normalized to the expected control per spray mass, where the control and each sprayed sample were measured in triplicate. HPLC results validate the UV-vis measurements, reporting efficiencies at $100 \pm 14\%$. The UV-vis and HPLC measurements were not statistically different, indicating that the efficiency determination method was validated (P-value = 0.9449). Pairwise comparison was conducted using an unpaired, two-tailed student t-test. **c** Gel EP results of (Lane 1) GLS-6150 control, (Lane 2) 1 kb ladder, and (Lane 3) sprayed GLS-6150 target.

statistically significant when compared to the UV-vis measurements ($104 \pm 10\%$) with a P-value of 0.3907. Thus, this enzymatic assay further confirms the validity of using rhodamine as an internal standard for UV-vis efficiency measurements.

### HPLC results of sprayed GLS-1027

HPLC was used with the immunomodulator, GLS-1027, to validate the UV-vis efficiency measurement approach. HPLC results demonstrated that GLS-1027 did not experience any significant molecular or chemical changes after being subjected to ESD. Retention times for sprayed targets were the same as the control (Fig. 5b). The profile of the targets indicates one and only one peak is present which follows the results of the control. Thus, no decomposition occured to the immunomodulator when deposited using ESD. No significant differences were observed between peak area measurements from control and sprayed targets as the sprayed targets reported $104 \pm 12\%$ recovery of the material. Finally, the efficiency results for GLS-1027 from HPLC were nearly identical to those measured with UV-vis ($99 \pm 18\%$) which further confirms the appropriate use of UV-vis for efficiency measurements (P-value = 0.9449) (see Supplementary Fig. 6).

### Gel electrophoresis results of GLS-6150

Agarose gel electrophoresis (EP) was performed on the DNA vaccine samples. The concentration of these samples was measured using the NanoDrop to validate the UV-vis measurements (P-value = 0.1591). GLS-6150 contains 4 different plasmids of sizes 3.6, 3.8, 4.4, and 5.1 kb. In

Fig. 5c, the GLS-6150 control is seen in lane 1 with the 4 plasmids present in both the open-circular conformation (5–8 kb) and the supercoiled conformation (2–3.5 kb). In lane 3, the sprayed sample is seen showing similar locations to the control in addition to maintaining both circular conformations post-processing. This indicates that through ESD, the DNA plasmid structures are not interrupted in the process of achieving high deposition efficiencies. However, future work is necessary to determine if ESD affects the ratios of the open-circular and supercoiled forms.

### Pierce 660 nm protein assay efficiency results of BSA

The concentration of BSA targets were measured using the NanoDrop to determine deposition efficiency. To validate these measurements further, Pierce 660 nm assay was performed on the same BSA targets. Using a standard curve generated from stock BSA solutions, the concentration of the experimental samples was determined, averaging a deposition efficiency of $107 \pm 11\%$. When compared to the NanoDrop measurements, the Pierce assay results were not statistically significant, yielding a P-value of 0.2401, thus confirming the deposition efficiencies of our BSA targets.

### Protein activity post-spray

In addition to spraying BSA, a trehalose-stabilized horseradish peroxidase complex was sprayed to test viability after ESD. Despite achieving efficiencies at $111 \pm 15\%$, electrospraying HRP dramatically decreased the functional activity of the enzyme, unlike GLS−1027 and GLS-6150.

Enzyme-linked immunosorbent assay (ELISA) results showed activity less than $4 \pm 0.06\%$ of the expected activity for the mass of protein sprayed. We suspect that the physical shear of the protein in the selected solvent blend via ESD decreased the HRP activity. This is supported by the fact that the HRP was observed to agglomerate over time in the syringe, indicating that, in this case only, the rhodamine and HRP efficiencies may have become decoupled. As discussed above, Morozov and Morozova demonstrated that it is indeed possible for proteins, specifically alkaline phosphatase, to be electrosprayed and still maintain high activity through an optimized protocol dependent on solvent formulation and the spray voltage and current used[12]. Due to the fact that they employed pure water in their formulations, we propose that these sprays may have occurred outside of the cone-jet mode, leading to both more favorable solvent and reduced shear conditions. Water's high surface tension makes it incompatible with the cone-jet spray without additional experimental modification, such as the use of an inert sheath gas.[45] Stabilization approaches are particular to the specific biomolecule; however, at this time, the requirements for high efficiency in the explored solvent system do not allow for retention of protein activity. Further solvents or protection mechanisms will need to be investigated should this be a requirement of the particular application.

## Comparison to other coating methods

It is important to consider the above results in the context of the dominant existing methods for precision soft coatings, specifically, dip coating, spin coating, and printing. Dip coating is currently a ubiquitous approach for coating MNAs and other surfaces for medical coatings in the range of 0.01–10 µm. To achieve quality dip coating deposition, several process parameters need to be considered[46]. Most relevantly, the capillary number of the coating fluid determines the fluid entrainment, with more viscous solutions and higher dipping rates leading to thicker films[31,47]. Thus, accurate dip coating requires a high degree of both mechanical and fluid formulation accuracy. Further, it is important to consider that thick films, particularly arising from high viscosity solutions, require more time for the coating to dry and may lead to drainage effects where the material may accumulate at the bottom of the substrate[46]. There is also inherent waste in dip coating from unutilized bath material, and the frequent insertion and removal increases the chances for fouling. Efficient ESD, by contrast, works exclusively with low viscosity solutions and can utilize nearly all the spray solution with the thickness controlled by spray time. Two advantages of dip coating over ESD, however, are that dip coating does not have the activity concerns reported here and the rate of dip coating deposition increases with target surface area while ESD is directly proportional to spray time. This said, ESD, especially when using self-limiting materials, can be employed for geometries that are much more complex than those that would be compatible with dip coating, such as foams[7]. Spin coating is by far the most precise method available for the deposition of micro-/nano-scopic thin films ranging from 0.01–200 µm film thickness[48]; however, it is inherently wasteful, removing greater than 90% of the utilized material. This waste becomes even greater when spatial control is desired, necessitating lift-off that removes any unpatterned surface and requires several lithographic processing steps. Further, spin coating is impractical on surfaces with anything beyond limited roughness (feature aspect ratio of approximately 0.1). However, it has the highest potential for uniformity, and further development of SLED will be necessary to target specific application areas where materials waste is a prime concern. Ink or electrohydrodynamic jet printing can target specific portions of a target with sub-micron spatial resolution through the use of piezoelectric stages; however, this accuracy comes at the cost of it being a serial process requiring precision positioning equipment, which cannot match the deposition rate of either dip or ESD coating. Further, geometries above "2.5-D" complexities cannot be targeted by

the line-of-sight nature of the jet, and capillary and gravity flow effects can lead to coating unwanted portions of the substrate[31]. Jet printing additionally requires careful formulation of the print solutions and their evaporative/wetting properties to avoid nozzle clogging and to achieve a uniform printed spot/line. Indeed, much as with the electrode templating shown in the T3 result, there is a large potential for ESD and jet printing to be used together with an optimal, conductive jet ink serving as the template for ESD.

## Discussion

By altering the charge landscape of the spray region, we show that ESD is a viable technology to coat MNAs and, more generally, for coating micro-scaled objects that are smaller than the spray plume. Various materials, ranging from sugar molecules to bioactive molecules, were sprayed onto MNAs, achieving deposition efficiencies at or around 100%. Our work demonstrates that, in addition to high deposition efficiencies, materials can be sprayed without disturbing the quality or decomposing the sprayed material for small molecules. This study shows that the capabilities of ESD are not limited to coating large targets and can be implemented for micro- and nano-fabrication applications where materials cost is a concern. We anticipate that future work will expand the range of compatible materials and the delivery rate of this high efficiency approach.

## Methods

### Materials

D-(+)-trehalose dihydrate ≥99%, PEGDA (Mn = 4,000), BSA aqueous solution (20 mg mL$^{-1}$), HRP Type VI ≥250 units (mg solid)$^{-1}$, Orgacon™ dry re-dispersable PEDOT:PSS pellets, rhodamine B ≥97%, MilliQ water, anhydrous 200 proof pure ethanol, 1-step Ultra 3,3′,5,5′-Tetramethylbenzidine (TMB)-ELISA substrate solution, and Pierce 660 nm protein assay reagent were obtained from Sigma Aldrich. Kollidon 12 PF (PVP) was obtained from BASF. GLS-6150 (Lot VGX-6150.13A001) and GLS-1027 (Lot 19AK0183A) were obtained from GeneOne Life Science, Inc. Sulfuric acid was obtained from Fisher Scientific. AZ400K developer (potassium-based buffered alkali solution) was obtained from MicroChemicals. PicoPure water was obtained from Hydro Service and Supplies.

### Solution preparation

PEDOT:PSS solutions were prepared by dialyzing dry-redispersable PEDOT:PSS pellets in water over 5 days, ending with a final concentration of 20 mg mL$^{-1}$. The dialyzed solution is further diluted to 10 mg mL$^{-1}$. For PEDOT:PSS and BSA, 100 µL of 10 mg mL$^{-1}$ solutions in water were mixed with 400 µL of pure ethanol and were sprayed onto 1 cm$^2$ silicon chips for ease of re-dispersion. All other solutions were sprayed onto microneedle substrates and prepared by mixing 100 µL of 10 mg mL$^{-1}$ solutions in water (or 200 proof ethanol in the case of GLS-1027), 10 µL of 25 µg mL$^{-1}$ rhodamine B, and 450 µL of pure ethanol.

### Microelectronic pattern fabrication

Parylene-C (5 µm thick) was deposited via chemical vapor deposition (CVD) using the SCS Labcoter®2 Parylene Deposition System PDS 2010 onto an A174 silane adhesion-promoter pre-treated 4-inch borofloat 33 glass wafer (University Wafer). The wafer was then treated for 60 s in an O$_2$ plasma treatment system (March PX-250) using 100 W at 80 mTorr to roughen the parylene layer. The patterning of the titanium/platinum (Ti/Pt) traces were achieved via photolithographic definition of the chip geometry. Briefly, a 3 µm photoresist (Kayaku Advanced Materials, Shipley S1818) layer was spun and exposed to define the patterns on the parylene-C coated wafer using the EVG620 UV Lithography System. The wafer was then treated again for 60 s with O$_2$ plasma using 100 W at 80 mTorr). A 200 nm thick Ti/Pt layer was

deposited using physical vapor deposition (Kurt J. Lesker PVD75) and released by lift-off in acetone bath.

## Electrospray setup

The electrospray setup involves a syringe pump (Harvard Apparatus 11 Plus), one negative high-voltage power supply (Acopian Power Supply, N012HA5), two positive high-voltage power supplies (Acopian Power Supply, P012HA5), a stainless-steel needle (SAI Infusion, 20 gauge, 0.5"), a steel guard ring (4 cm outer diameter, 2 cm inner diameter), 2 mm long stainless-steel microneedles in a $4 \times 4$ array, and a humidity and temperature-control environmental chamber (Electro-Tech Systems, Inc.). Power supplies were controlled using a custom LabVIEW script (National Instruments Corp. version 2020). The chamber had a controlled humidity ranging from 15–25% RH, and the temperature ranged from 24–27 °C. The spray solution was loaded into 1 mL Luer Lock syringes with an inner diameter of 4.78 mm. The microneedles were placed on an aluminum holding block where the microneedle array and the holding block were grounded. A positive voltage was applied and adjusted as needed on the syringe needle and the guard ring. A negative pre-spray using 200 proof ethanol was sprayed before each target. All conductive material within the chamber was insulated with 1 layer of 2 mil Kapton polyimide tape, including the guard ring and the holding block. Microneedle arrays were sonicated with detergent and water for cleaning.

## Experimental parameters

A negative-polarity ethanol spray (negative potential ranging from 5–6 kV) was used for a minimum of 5 min prior to substrate collection. The negative pre-spray treatment was applied every 3 h or when spray instabilities arose. Sprays were stabilized using the primary voltage with a range of 6–8.5 kV at a constant spray distance of 2 cm to the ring and 4 cm to the target. The ring voltage was held at 0.41 kV. Ambient humidity was regulated between 15–25% RH. All sprays occurred at a flow rate of 0.1 mL h$^{-1}$, corresponding to a mass delivery rate of 180 µg h$^{-1}$ of the payload material and 45 ng h$^{-1}$ of tracer. For efficiency calculations, sprays were collected for 30 min, resulting in 90 µg of material and 45 ng of tracer.

## Apparent deposition efficiency calculation

PEDOT:PSS and BSA substrates were dipped into 600 µL of water and measured using the UV-vis and BSA protein A280 function, respectively, of the Thermo Scientific NanoDrop 2000C. Measurements were taken at 260 nm, and no baseline correction was used for the measurements. See Supplementary Fig. 7 for calibration curve information of PEDOT:PSS.

For all other materials, needles or chips were dipped into 600 µL of water, or AZ400K Developer for GLS-1027, for 2 min until all material is dissolved off the needles. The solution is then analyzed with a Jasco 770 UV-vis spectrophotometer. The results were then background subtracted using a Gaussian fit to extract the rhodamine peak using a custom MATLAB script (MathWorks, Inc. version R2022a), which was compared to a standard calibration curve generated in the same solvent.

## Trehalose assay

Trehalose samples were prepared by mixing 100 µL of 10 mg mL$^{-1}$ solutions in water and diluted with 400 µL of 200 proof ethanol. The solution was sprayed for 30 min onto a $4 \times 4$ MNA ($n = 3$). The needles were eluted into 300 µL of PicoPure water and analyzed using an enzymatic trehalose assay kit obtained from Megaenzyme Ltd. (Bray, Ireland).

## HPLC analysis

HPLC measurements were performed by MedChem 101, LLC (Plymouth Meeting, PA) using the Agilent 1260 Infinity II. The column used is a Porashell 120 C18 (EC C18 4 µm) $150 \times 4.6$ mm at 20 °C. The mobile

phase consisted of 45 vol% acetonitrile + 0.05 vol% trifluoroacetic acid and 55 vol% deionized water + 0.05 vol% trifluoroacetic acid. The samples were flowed at 0.8 mL min$^{-1}$ for 8 min using a sample volume of 4 µL. Samples were detected using UV at 254 nm. HPLC measurements were isocratic.

GLS-1027 solutions were prepared by mixing 100 µL of 10 mg mL$^{-1}$ solutions in 200 proof ethanol and diluted with 400 µL of 200 proof ethanol. The solution was sprayed for 90 min onto a $8 \times 8$ MNA ($n = 3$). The needles were then eluted in 500 µL of 200 proof ethanol and given to MedChem 101, LLC, for HPLC analysis.

## Hydrogel preparation

Hydrogel solutions were made by mixing 100 µL of 10 mg mL$^{-1}$ PEGDA in water with 3.6 µL of 10 mg mL$^{-1}$ DMPA in 200 proof ethanol. 396.4 µL of 200 proof ethanol was added to this mixture for a total solution volume of 500 µL. The solution was then sprayed for 30 min onto a $1 \times 4$ MNA for ease of visual purposes. After spraying, the coating was densified using a humidifier. When the coating was densified and visually clear, the coating was cured for 5 min using a Dymax Light-Welder PC-3 UV Spot Lamp. The MNA was then dipped in water mixed with rhodamine B and imaged using a Dinolite USB Camera under a UV light.

## Gel EP

Samples were loaded onto Bio-Rad 1% TAE Mini ReadyAgarose Precast Gel 12-well gels. For the DNA ladder, New England Biolabs 1 kb DNA Ladder was used. 10 µL of stock GLS-6150 at 41.7 µg mL$^{-1}$ was loaded. MNAs sprayed with GLS-6150 were eluted into 600 µL of water. 10 µL of this sample was loaded into the well. All samples used New England Biolabs Gel Loading Dye, Purple (6X). The gel was run for 80 min at 80 V in 1X TAE buffer. The gel was then stained for 40 min in 1:10,000 SYBR Safe DNA Gel Stain:1X TAE buffer. Syngene PXi Gel Imaging system was used to image the gel after staining.

## ELISA

HRP activity was measured using an ELISA reagent in 96-well polystyrene tissue culture plates (VWR, Radnor, PA). HRP Type IV was dissolved in molecular biology grade water at 1 wt% as a standard. HRP standard was diluted to 10 ng mL-1 (1:1,000,000) in PicoPure water, and a standard curve was generated from 0.31 ng mL$^{-1}$ to 10 ng mL$^{-1}$ using serial dilutions with PicoPure water. Two experimental HRP substrates from collected electrospray were prepared at theoretical concentrations of 20 µg mL$^{-1}$ and further diluted to 1:250 in water. HRP solutions were added to the plates with an equal volume of the 1-step Ultra TMB-ELISA substrate solution. Plates were incubated for 5 min at room temperature, and 50 µL of 2 M sulfuric acid was added to stop the reaction. Absorbance was measured at 450 nm and 540 nm on a Tecan Infinite M200 Pro plate reader (Tecan, Männedorf, Switzerland), and the background (540 nm) was subtracted from the 450 nm signal. Samples ($n = 4$) were averaged for each condition, and experimental active HRP concentrations were determined from the standard curve in Supplementary Fig. 8.

## Pierce 660 nm protein assay

BSA protein standards and targets were prepared by combining 10 µL of BSA solution with 150 µL of the reagent and aliquoting into separate wells. These solutions were then mixed gently and allowed to incubate at room temperature for 5 min. Absorbance was measured at 660 nm on a Tecan Infinite M200 Pro plate reader. Samples were measured $3 \times 3$ and averaged for each condition. Experimental BSA concentrations were determined from the standard curve in Supplementary Fig. 9.

## Finite element model

The FEM model was performed using COMSOL 6.0. Additional details of the simulation can be found in the Supporting Information.

## Reporting summary

Further information on research design is available in the Nature Portfolio Reporting Summary linked to this article.

## Data availability

The data supporting the conclusions of this study are included the article and the supplementary information files. The UV-vis datasets generated in this study have been deposited in the Figshare database (https://doi.org/10.6084/m9.figshare.23257496)[49]. The HPLC datasets generated in this study have been deposited in the Figshare database (https://doi.org/10.6084/m9.figshare.23600460.v1)[50].

## Code availability

The MATLAB and LabVIEW scripts and COMSOL code used for this study have been deposited in the Figshare database (https://doi.org/10.6084/m9.figshare.23609133.v1)[51].

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

## Acknowledgements

The authors acknowledge the financial support from GeneOne Life Science via a sponsored research agreement awarded to J.P.S.

## Author contributions

J.P.S., L.L., and S.H.P. conceptualized the experimental design. S.H.P., L.L., D.D., E.O.L., M.A., E.T.D., and R.Z. performed the experiments. JPS performed the finite element analysis. S.H.P. and J.P.S. wrote the manuscript. S.H.P., J.P.S., J.N.M., H.L., D.I.S., J.D.Z., and J.W.S. revised and reviewed the manuscript.

## Competing interests

Author J.N.M. is an employee of GeneOne Life Science. As such, J.N.M. receives salary and benefits, including ownership of stock and stock options. All other authors declare they have no competing interests.
