## [Peer Review File · Nature Communications]

nature portfolio

Peer Review FileReviewer comments, first round

Reviewer #1 (Remarks to the Author):

This manuscript describes an experimental scheme to obtain efficient electrospray deposition onto small targets (less than the size of the unaltered plume diameter).

The study describes several noteworthy results, namely nearly 100% deposition efficiency onto microneedle array targets. The authors deposit a range of materials, including small molecule dyes, polymers, and proteins. The range of materials is significant because it suggests the approach is generalizable.

The impact of the work is lessened considering that nearly 40% efficiency was obtainable without the modifications described in the paper.

The novelty of the work is modest because the improvements, such as the third electrode and use of insulating masks, have been used elsewhere as cited by the authors. The authors contribution is to combine various known improvements into a single well-designed and well-characterized experimental scheme.

I am not convinced that the use of rhodamine as a tracer molecule was adequately validated. The assumption is made that rhodamine deposition efficiency matches that of the other materials, despite very different sizes, chemical/physical properties, and concentration regimes. An experiment where the payload deposition efficiency is measured directly and compared to the tracer is needed.

The authors state in the abstract that the coating is "uniform," but do not demonstrate uniformity of coating on the deposition targets. Perhaps the authors meant "repeatability," which they did demonstrate as evidenced by the reduced standard deviations in, e.g., figure 2.

I suggest avoiding the use of the term "sample" because it is confusing what that encompasses. Perhaps "target" or "deposition substrate" is more clear.

Reviewer #2 (Remarks to the Author):

This manuscript reports on the use of electrospray deposition to deliver materials to surfaces smaller than the spray plume. This is achieved by modulating the electric charge/field in the vicinity of the target. Bioactive compounds (among other materials) are deposited and their viability is confirmed. Very high deposition efficiencies are achieved.

This is a well-written manuscript and a great contribution on the electrospray deposition of materials. As noted by the authors, while electrospray has several unique advantages, the deposition efficiency has been shown to be inherently low. Manipulating the local electric field (as shown here) is a great option to enhance the deposition efficiency; and novel insights are provided in this research. A strength of the work is the variety of model materials that were evaluated for deposition. The compelling evidence that the viability of the bioactive materials is preserved after electrospray will potentially broaden the appeal of this process.

I am curious about the negative pre-spray process. Presumably this is performed to eliminate any residual charge in the vicinity of the target (which may prevent deposition even on conductive targets and could destabilize the spray). It would seem that the overall process is intended to be conducted in a periodic manner - where the negative spray is used for some time (to create a "blank slate") followed by the positive polarity spray to deposit material (then repeat). But what sort of spray times are necessary? What would be the minimum spray time required to create this blank slate? And how long can the positive mode be used (depositing material) before the negative mode is needed again?

I note that the focusing ring is also coated in polyimide (Kapton) - but does not seem to be subject to the negative polarity spray. Is this correct? Will it not also accumulate charge and destabilize the spray?

Can the authors comment on the flow of current to the extractor ground? In the reference to Kingsley et al. (Ref 10), it was found that the current (for the image charge) was constant over a fairly long time. This implied that charge continuously accumulated on the insulator *or* a steady state had been reached, where the arriving charge was balanced by the decay. What do the authors think occurs in their system? Does the charge reach a fixed value on the surface of the Kapton (acting like a capacitor) - or does it decay, in both positive and negative mode? And to this end - does grounding the extractor permit more charge to accumulate on the Kapton compared to the floating condition?

Can the authors comment on any difference in the processing for positive and negative mode? Were comparable potentials used (magnitudes)?

A small point: can the authors confirm what is meant by "unwanted regions" on Page 3?

Reviewer #3 (Remarks to the Author):

This work by Sarah Park, et al. presents a novel method for electrospray deposition on materials smaller than the plume of the plume. The authors have successfully overcome previous limitations of spraying on such small surfaces, and the range of target surfaces and spray materials used in this study was substantial. The paper provides strong evidence for the method's efficiency and the double-validation of the UV-vis results via HPLC. The authors also present weaknesses of the method. For example, the ELISA results reported in the paper are unfavorable for the method's application in keeping protein activities. While the authors provide a rational explanation for this outcome, the reliance on literature alone feels insufficient. Further experimentation will be helpful. In general, the results obtained within this work are promising, and the figures and photographs provided were efficient in demonstrating what was being discussed. Therefore, with minor revisions to address the concerns below, I support this work for publication.

1. Line 71-74: the previous sentence introduces three different chip substrates, but the following sentence, through the use of "former" and "latter," only serves to describe two, and it is ambiguous what is being referred to in these descriptions due to this discrepancy.
2. Line 142: deposition efficiency of trehalose sprays is described to be "essentially 100%," although Figure S3 makes this seem like an overgeneralization, as the 9.3 ug/needle bar is closer to 75% than 100, and the range from the lowest efficiency to the highest looks to be reasonably broad (~30%), so combining all these outcomes as essentially the same result seems inaccurate. More hedged language or a more nuanced description of the results may be more accurate.
3. Please define abbreviations before using them. This includes HPLC (line 283 and Fig 5 caption at line 263), and ELISA (318).
4. Results section 3.5 (lines 305-330) discusses how HRP activity was less than 4% of what would be expected. Although the previous literature mentioned has established that proteins can maintain activity after ESD spray, the inability to meaningfully recreate those results here presents concerns. The protein activity reported in literature should be at least somewhat recreated experimentally before stating the method's potential applications for protein spray, as the discrepancy between the experiment and literature here undercuts the method's viability for this application.
5. Methods section 5.7 (lines 446-456): please provide the HPLC gradient used.

In the following response, Reviewers' comments have been presented in italics and our replies in normal text. We thank the Reviewers for their thorough consideration of our manuscript.

Reviewer #1: This manuscript describes an experimental scheme to obtain efficient electrospray deposition onto small targets (less than the size of the unaltered plume diameter). The study describes several noteworthy results, namely nearly 100% deposition efficiency onto microneedle array targets. The authors deposit a range of materials, including small molecule dyes, polymers, and proteins. The range of materials is significant because it suggests the approach is generalizable.

The impact of the work is lessened considering that nearly 40% efficiency was obtainable without the modifications described in the paper.

The novelty of the work is modest because the improvements, such as the third electrode and use of insulating masks, have been used elsewhere as cited by the authors. The authors contribution is to combine various known improvements into a single well-designed and well-characterized experimental scheme.

The authors appreciate the Reviewer's comments. While the use of certain modifications in the electrospray setup have been established prior to this work (e.g.: the focus, or guard, ring and the insulative mask), here, we aim to characterize the impact of these modifications, in addition to other modifications our work adds, specific to coating substrates much smaller than the spray plume. To our knowledge, studies achieving high deposition efficiency at this scale seem to be limited at this time, and further, studies that do exist show inconsistent effects of the modifications.

I am not convinced that the use of rhodamine as a tracer molecule was adequately validated. The assumption is made that rhodamine deposition efficiency matches that of the other materials, despite very different sizes, chemical/physical properties, and concentration regimes. An experiment where the payload deposition efficiency is measured directly and compared to the tracer is needed.

While true that rhodamine B is on the lower end of the molecular weights studied, we posit that any materials that remain fully dissolved up to the point of spray must have similar efficiency as there is no mechanism for separation. We do appreciate, however, that dissolution in the Taylor cone is difficult to validate without evidence of excessive residues, etc. To this end, we have included confirmatory measurements via several techniques, including high performance liquid chromatography (HPLC), Nanodrop (a second UV-vis system) measurements, and a Pierce 660 nm assay. These efficiency measurements were valid in comparison to the UV-vis measurements using rhodamine B for a variety of payload materials. When comparing the efficiency measurements of GLS-1027, the small immunomodulator molecule, there was no statistical significance observed between the UV-vis measurements and the HPLC measurements (P-value = 0.9449) despite rhodamine (479 g mol^{-1}) being much larger in MW than GLS-1027 (205 g mol^{-1}). In the case of the Nanodrop, GLS-6150 (3.6-5.1 kb) was measured, demonstrating no statistical

significance in comparison to the rhodamine UV-vis measurements (P-value = 0.2957). Further, the Nanodrop was used to obtain consistent efficiency measurements for PEDOT PSS. To add to this confidence, we have conducted an additional trehalose colorimetric assay (and added the associated methods), which also indicated no statistical significance between the trehalose assay measurements and the UV-vis measurements, thus validating the rhodamine UV-vis approach once more.

To summarize these results, an additional figure has been added to the Supplementary Materials (new Supplementary Fig. 5) that compares all UV-vis measurements to each of the various confirmatory methods.

Supplementary Figure 5. Apparent deposition efficiencies measured using a UV-vis approach with rhodamine B as an internal standard (n = 30) compared to other methods of measurement using trehalose assay kit (n = 3), HPLC (n = 3), Nanodrop UV-vis (n = 12), and Pierce 660 nm assay (n = 6).

The authors state in the abstract that the coating is "uniform," but do not demonstrate uniformity of coating on the deposition targets. Perhaps the authors meant "repeatability," which they did demonstrate as evidenced by the reduced standard deviations in, e.g., figure 2.

The authors appreciate the suggestion and have made the following change highlighted on Page 1.

I suggest avoiding the use of the term "sample" because it is confusing what that encompasses. Perhaps "target" or "deposition substrate" is more clear.

The authors appreciate the suggestion and have changed the word "sample" to "substrate" or "target" throughout the manuscript where appropriate.

Reviewer #2: *This manuscript reports on the use of electrospray deposition to deliver materials to surfaces smaller than the spray plume. This is achieved by modulating the electric charge/field in the vicinity of the target. Bioactive compounds (among other materials) are deposited and their viability is confirmed. Very high deposition efficiencies are achieved.*

This is a well-written manuscript and a great contribution on the electrospray deposition of materials. As noted by the authors, while electrospray has several unique advantages, the deposition efficiency has been shown to be inherently low. Manipulating the local electric field (as shown here) is a great option to enhance the deposition efficiency; and novel insights are provided in this research. A strength of the work is the variety of model materials that were evaluated for deposition. The compelling evidence that the viability of the bioactive materials is preserved after electrospray will potentially broaden the appeal of this process.

I am curious about the negative pre-spray process. Presumably this is performed to eliminate any residual charge in the vicinity of the target (which may prevent deposition even on conductive targets and could destabilize the spray). It would seem that the overall process is intended to be conducted in a periodic manner - where the negative spray is used for some time (to create a "blank slate") followed by the positive polarity spray to deposit material (then repeat). But what sort of spray times are necessary? What would be the minimum spray time required to create this blank slate? And how long can the positive mode be used (depositing material) before the negative mode is needed again?

The authors appreciate the positive feedback and are happy to elaborate more on the negative pre-spray process. We applied the negative polarity ethanol (EtOH) pre-spray for approximately 5 min prior to substrate collection. After doing so, the negative spray would be employed between every 3 hours of sample collection or when the Taylor cone experienced instabilities. We have added these details on Page 11 under experimental parameters.

“A negative-polarity EtOH spray (negative potential ranging from 5-6 kV) was used for a minimum of 5 min prior to substrate collection. The negative pre-spray treatment was applied every 3 h or when spray instabilities arose.”

I note that the focusing ring is also coated in polyimide (Kapton) - but does not seem to be subject to the negative polarity spray. Is this correct? Will it not also accumulate charge and destabilize the spray?

Thank you for noting this – indeed, the focus ring is also subjected to the negative polarity spray, and Figure 1 has been edited to reflect as such on Page 3.

Figure 1. **Schematic of the electrospray setup.** **a** Schematic and **b** photograph of an ESD experiment where a spray plume directed at a grounded target is generated from a solution reservoir held at high voltage. White dashed lines provided as a guide to the eye of the plume. **c** Schematic of the spray system and process, highlighting different enhancements (denoted ‘EX’). In stage 1, a negative-polarity ethanol spray (E1) is sprayed directly at a large extractor ground (E2) which is coated in insulating Kapton tape (E3). While the focus ring is in place during this treatment, no clip is applied and thus it is not electrified or grounded. Then in stage 2, a grounded target with an insulating mask (E4) is placed on the extractor ground. It is then sprayed by the spray solution at positive polarity which is stabilized by a focus ring (E5). The ring, and all other proximate metal surfaces, is also coated in insulating tape.

*Can the authors comment on the flow of current to the extractor ground? In the reference to Kingsley et al. (Ref 10), it was found that the current (for the image charge) was constant over a fairly long time. This implied that charge continuously accumulated on the insulator *or* a steady state had been reached, where the arriving charge was balanced by the decay. What do the authors think occurs in their system? Does the charge reach a fixed value on the surface of the Kapton (acting like a capacitor) - or does it decay, in both positive and negative mode? And to this end - does grounding the extractor permit more charge to accumulate on the Kapton compared to the floating condition?*

With respect to the flow of current to the extractor ground, we hypothesize that the charge accumulation reached a fixed value or was decaying too slowly to observe any effects during our spray times. We suspect this is occurring as the extractor ground does not seem to accumulate more charge/material during our spray times and our polyimide tape is much thicker (4 mil) in comparison to the polyimide film used in Kingsley et al. However, if we were to spray for much longer, this may no longer be the case as the field lines are more directed towards the target in the presence of the extractor ground, those indicating charge accumulation on the polyimide tape surface. Additionally, Yingshun et al (Journal of Power Sources, Volume 366, 2017). has shown that polyimide coatings are able to withstand high volume charge-discharge cycles while maintaining a relatively constant capacity retention for battery applications at ambient conditions. Thus, this further suggests that polyimide film requires a longer amount of time to pass before observing a decay in charge accumulation.

As to whether grounding the extractor results in more or less charge accumulation, we did not evaluate this in detail; however, our current results allow us to speculate. If there was no target present, the floating case would quickly saturate and redirect spray to whatever ground is available. Indeed, the spray stabilization would require the presence of such a ground and the floating target would receive spray only indirectly. By contrast, a grounded extractor lacking a target would allow for a more sustained spray to reach the Kapton coating. While a more complicated scenario, we anticipate that this would be the same for a spray conducted with a target—the grounded extractor both stabilizes the spray and allows for a greater amount of charge to accumulate on the extractor. This said, as evidenced by the efficiency numbers, this quantity must be *de minimis* with respect to the amount of charge reaching the target. Further, when several samples are sprayed, only the first spray would be expected to lose material to the extractor.

Can the authors comment on any difference in the processing for positive an negative mode? Were comparable potentials used (magnitudes)?

The negative potential was approximately 5-6 kV as that was the applied voltage needed in our setup to spray EtOH. The cone, however, Taylor cone-jet stability was not enforced and some drips occurred throughout the 5 min, mainly to disperse negatively polarized droplets of the solvent throughout the spray chamber (description of the pre-spray process seen on Page 11):

“A negative-polarity EtOH spray (negative potential ranging from 5-6 kV) was used for a minimum of 5 min prior to substrate collection. The negative pre-spray treatment was applied every 3 h or when spray instabilities arose.”

Additionally, no evidence was shown in our experimentation that in cone-jet mode, applying a negative or positive polarity did not cause any differences in or change the behavior of our sprays – though we did not study this thoroughly. Much of ESD literature has been sprayed with positive potentials, and, as a result, our work was conducted similarly but a future study specific to this work would be interesting to confirm our hypothesis. Our lab has published a manuscript in 2022 (Green et al. Appl. Polym. Mater., Vol. 4, 2022) in which negative potentials are used for their sprays showing the same behavior with regards to charge induced self-limiting thickness as positive polarity sprays; however, a different ESD setup is used, and thus we are hesitant to state that there is no difference in this work.

A small point: can the authors confirm what is meant by "unwanted regions" on Page 3?

The term “unwanted regions” refers to areas on our substrate that may attract spray deposition due to conductivity but otherwise would not be useful for its intended application. In the example of our microneedle array (MNA) geometry, the electrodes on the MNA are conductive and can

accumulate a coating (seen in Page 4, Fig. 2d) as a result. More clarity to the term was added on Page 3:

“... (e.g.: exposed electrodes on the MNA)...”

Reviewer #3: *This work by Sarah Park, et al. presents a novel method for electrospray deposition on materials smaller than the plume of the plume. The authors have successfully overcome previous limitations of spraying on such small surfaces, and the range of target surfaces and spray materials used in this study was substantial. The paper provides strong evidence for the method's efficiency and the double-validation of the UV-vis results via HPLC. The authors also present weaknesses of the method. For example, the ELISA results reported in the paper are unfavorable for the method's application in keeping protein activities. While the authors provide a rational explanation for this outcome, the reliance on literature alone feels insufficient. Further experimentation will be helpful. In general, the results obtained within this work are promising, and the figures and photographs provided were efficient in demonstrating what was being discussed. Therefore, with minor revisions to address the concerns below, I support this work for publication.*

1. Line 71-74: the previous sentence introduces three different chips substrates, but the following sentence, through the use of “former” and “latter,” only serves to describe two, and it is ambiguous what is being referred to in these descriptions due to this discrepancy.

The authors appreciate the kind support and suggested revisions request by the Reviewer. We have fixed the text to denote more clearly what is meant by former and latter on Page 2 and, instead, named the specific geometries to avoid ambiguity.

2. Line 142: deposition efficiency of trehalose sprays is described to be “essentially 100%,” although Figure S3 makes this seem like an overgeneralization, as the 9.3 ug/needle bar is closer to 75% than 100, and the range from the lowest efficiency to the highest looks to be reasonably broad (~30%), so combining all these outcomes as essentially the same result seems inaccurate. More hedged language or a more nuanced description of the results may be more accurate.

On Page 4, we have changed the word “essentially” to “approaching” such that we are able to account for these nuances within our data.

3. Please define abbreviations before using them. This includes HPLC (line 283 and Fig 5 caption at line 263), and ELISA (318).

Thank you for bringing this to our attention – we have since defined the abbreviations for HPLC (Page 1) and ELISA (Page 9).

4. Results section 3.5 (lines 305-330) discusses how HRP activity was less than 4% of what would be expected. Although the previous literature mentioned has established that proteins can maintain activity after ESD spray, the inability to meaningfully recreate those results here presents concerns. The protein activity reported in literature should be at least somewhat recreated experimentally

before stating the method's potential applications for protein spray, as the discrepancy between the experiment and literature here undercuts the method's viability for this application.

We appreciate the Reviewer's comments on this key point. We believe that the low activity may be a function of the high concentration of added ethanol that needs to be added to increase stability of the electrospray solution as well as the shear stress involved with the cone-jet spray mode. To address the Reviewer's concern, we have attempted to replicate the results from literature to the best of our experimental capability using the experimental details available in Morozov and Morozova, with the hypothesis, as stated in our manuscript, that they were spraying in a different mode. We reconfigured our setup to use similar spray conditions with a spray distance of 1 cm using a positive potential of 3 kV. Our syringe pump was unable to flow at $0.1 \mu\text{L min}^{-1}$, so the solution was flowed at the lowest possible flow rate at $0.15 \mu\text{L min}^{-1}$. The protein solution was sprayed at the same concentration of 0.20 mg mL^{-1} in water. Water is typically not considered to be a stable electrospray solution (Tang et al., Journal of Aerosol Science, Volume 25, 1994, we have also added this reference to the revised manuscript on Page 9 as follows:

“Water's high surface tension makes it incompatible with the cone-jet spray without additional experimental modification, such as the use of an inert sheath gas.⁴⁸”

Seen in Fig. R1a, the resulting sprays produced larger droplets using water as a solvent, whereas our ESD setup uses a 1:4 water:EtOH mixture as a solvent and applies a voltage between 6-8.5 kV, resulting in finer droplets that only deposits these larger droplets on the substrate in the event of unintended and infrequent instability in the cone.

The high concentration of EtOH can be expected to negatively affect the activity of HRP, which we have confirmed through TMB assay. This is seen in Fig. R1b where we measured the activity of HRP after exposure to 95% EtOH over time. Upon suspension in ethanol, the activity of the protein begins to drop, and at 3 h or longer there is already approximately a 50% decrease in activity. This decrease is before any further decrease due to shear stress occurring at the stable Taylor cone. Further, in Fig. R1c, the spray solution can be observed to be opaque in the syringe almost immediately upon making the solution, showing the precipitation of HRP as mentioned in the main manuscript text. This would lead to a caveat to the rhodamine efficiency measurements, where gravitational precipitation or residue formation of the protein could lead the rhodamine efficiency measurements to be inaccurate with respect to the protein solution concentration. It is anticipated that the three combined effects lead to the reported activity measurements in the manuscript; however, we feel that it would be a distraction to focus on these points in the current discussion, which is focused on efficiency, and hope that the Reviewer agrees that this could be a potential direction for future work.

Figure R1. **Replication of HRP experiments.** **a** Image of the substrate after spraying using the experimental conditions of Morozov and Morozova. The HRP solution was sprayed with rhodamine B such that fluorescent microscopy can be utilized to image the droplet sizes after deposition. **b** Activity of HRP in EtOH over time using a TMB assay. Measurements were taken for 0, 1, 2, 4, and 24 h time elapsed. **c** Image of 2 mg mL⁻¹ of HRP in 1:4 water:EtOH prior to spraying the solution.

5. *Methods section 5.7 (lines 446-456): please provide the HPLC gradient used.*

No HPLC gradient was used as all measurements were isocratic. This detail was added to the HPLC methods section on Page 12.

Reviewer comments, further round

Reviewer #1 (Remarks to the Author):

The authors have adequately addressed reviewer comments, in my opinion, and the manuscript is fit to be published.

My one objection is the line in the manuscript that states "the fact that proteins can retain activity in electrospray suggest that it may be possible to achieve ESD of active protein coatings that have a more targeted, higher efficiency with further study." In light of reviewer #3's comments on protein activity and the authors' response, I think this sentence is speculative and not supported by the data. I think the reasonable conclusion is that high efficiency deposition is not, in the current implementation, compatible with maintaining high enzyme activity.

Reviewer #2 (Remarks to the Author):

All of my comments have been suitably addressed. Thank you.

In the following response, Reviewers' comments have been presented in italics and our replies in normal text. We thank the Reviewers for their thorough consideration of our manuscript.

Reviewer #1: *The authors have adequately addressed reviewer comments, in my opinion, and the manuscript is fit to be published.*

My one objection is the line in the manuscript that states "the fact that proteins can retain activity in electrospray suggest that it may be possible to achieve ESD of active protein coatings that have a more targeted, higher efficiency with further study." In light of reviewer #3's comments on protein activity and the authors' response, I think this sentence is speculative and not supported by the data. I think the reasonable conclusion is that high efficiency deposition is not, in the current implementation, compatible with maintaining high enzyme activity.

The authors appreciate the Reviewer's input and have addressed the reviewers' concern on Page 13:

“...at this time, the requirements for high efficiency in the explored solvent system do not allow for retention of protein activity. Further solvents or protection mechanisms will need to be investigated should this be a requirement of the particular application.”

Reviewer #2: *All of my comments have been suitably addressed. Thank you.*

We appreciate the Reviewer's previous suggestions and consideration of our manuscript.